# Part 1: Multifractal analysis of wind turbine power and the associated biases

Jerry Jose[a], Auguste Gires[a], Yelva Roustan[b], Ernani Schnorenberger[c], Ioulia Tchiguirinskaia[a], Daniel Schertzer[a]

[a]*HM&Co, École nationale des ponts et chaussées | Institut Polytechnique de Paris, 77455 Champs-sur-Marne, France*
[b]*CEREA, École nationale des ponts et chaussées | Institut Polytechnique de Paris, EDF R&D, Île-de-France, France*
[c]*Boralex, Lyon, France*

Correspondence to: Jerry Jose (jerry.jose@enpc.fr)

## Abstract

The inherent variability in atmospheric fields, which extends over a wide range of temporal and spatial scales, also gets transferred to energy fields extracted off them. In the specific case of wind power generation, this can be seen in the theoretical power available for extraction as well as the empirical power produced by turbines. For modelling and analyzing them, it is important to quantify their variability, intermittency and correlations with other interacting fields across scales. To understand the uncertainties involved in power production, power outputs from four 2 MW turbines are analyzed (from an operational wind farm at Pay d'Othe, 110 km southeast of Paris, France) using the scale invariant framework of Universal Multifractals (UM). Their scaling properties were compared with power available at the same location from simultaneously measured wind velocity.

While statistically analysing the turbine output, the rated power acts like an upper threshold resulting in biased estimators. This is identified and quantified here using the theoretical framework of UM and validated using numerical simulations. Understanding the effect of instrumental thresholds in statistical analysis is important in retrieving actual fields and modelling them, more so, in wind power production where the uncertainties due to turbulence are already a leading challenge. This is expanded in Part 2 where the influence of rainfall in power production is studied across scales using UM and joint multifractals.

*Keywords:*
wind, multi fractal, wind power, wind turbine

## 1. Introduction

In the increasing global transition towards renewable and carbon-neutral energy, wind power is extremely attractive as it has some of the lowest carbon emission in life cycle assessment (Li et al., 2020; Guezuraga et al., 2012; Wiser et al., 2011). The levelized cost of energy (LCOE, cost including building and operation) has also decreased drastically in past decades for both offshore and onshore wind power (80 % since early 1980, and further 30 % in past 5 years) giving it better economic value (Beiter et al., 2021). However, wind is a fluctuating field and owes its generation mainly to uneven heating of the earth's surface by solar radiation and the pressure gradients generated from it. Further, atmospheric turbulence makes the characterization of the field a difficult task (with governing Navier-Stokes equations still remaining unsolvable). The small-scale fluctuations and intermittency in wind are transferred to power produced; this is further complicated by the fact that wind turbine hubs are located in the atmospheric boundary layer. In addition, an improved understanding of turbulence is identified as one of the leading challenges in the field of wind power by experts (van Kuik et al., 2016). When it comes to the working of modern turbines, one way to account for wind variations is through variable speed turbines and adaptive torque control enabling maximum power capture. However, the commonly used parameter for control, 'turbulence intensity' (standard deviation of wind speed divided by mean wind speed over 10 min) cannot fully capture the behaviour (see non-Gaussian behaviour of wind velocity in Fig. 1), and is too coarse to represent the variability (active torque controls should responsive down to a few seconds). Further, this doesn't consider any effect of rain that could get transferred to loads on turbine (Johnson, 2004).

To understand the complex effect of turbulence on power production, along with access to high-resolution data, an appropriate theoretical framework is required to characterize intermittency at all scales of measurement. The scale invariant multifractal framework of Universal Multifractals (UM), which is widely used to study the variability in geophysical fields, can be used to characterize this complexity (Schertzer and Lovejoy, 1987, 1997). Using the framework of UM, Fitton et al. (2011, 2014) studied scaling behaviour and multifractal properties of wind velocity and torque fluctuations in wind farm test sites (in Germany and Corsica), and made a case for multifractal modelling of atmospheric turbulence. Multifractality of wind speed and aggregate wind farm power was illustrated in Calif and Schmitt (2014) where the coupling between both fields were examined.

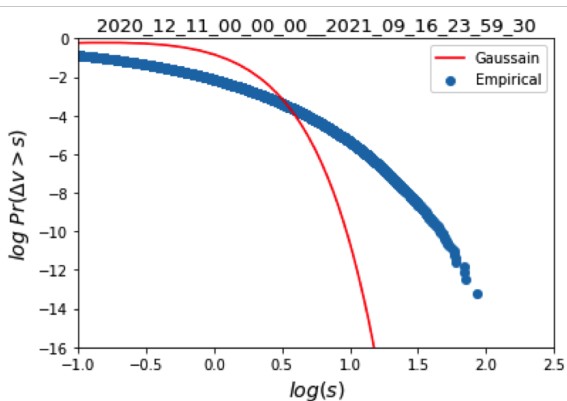

Figure 1: Log-log plot of exceedance probability, $Pr(\Delta v > s)$, of only positive horizontal velocity increments, $\Delta v(\tau) = v(t + \tau) - v(t)$, during (Dec 2020 to July 2021, at 1 Hz from location 1 of RW-Turb meteorological mast) along with a Gaussian distribution to illustrate latter's inadequacy. $s$ is a threshold of intensity and $\tau$ here is 15 s.

In light of the scientific perspectives mentioned so far, here we try to characterize the small-scale fluctuations in wind power production using data from an operational wind farm at Pays d'Othe, 110 km southeast of Paris, France. However, while analysing the variability of field using statistical methods, presence of instrumental limits in data can introduce biases. For example, the effect of instrumental lower threshold is discussed in (Jose et al., 2021) within the framework of UM analysis using atmospheric extinction coefficient ($\sigma_e$) as the field. Similarly, there is also the bias from presence of zeroes in data (Gires et al., 2014). Both of these biases are present in statistical analysis of empirical power from wind turbine since the turbine is designed to work at a rated power (here, 2 MW) and can provide zero or negative power (more consumption than production). The major aim of this paper is to highlight these biases encountered during multifractal analysis and its influence on direct statistical analysis of turbine power. For the theoretical aspect only the effect of upper threshold, which is not yet addressed in there literature, is considered here to avoid complexity. More analysis is intended for a follow-up paper where the influence of rain in wind power production is examined along with the coupling of power as a field with other atmospheric parameters. The details of data collection and quality are presented in the upcoming section on data and methods; the second part of this section briefly recapitulates the framework of UM. The biases encountered in the analysis of turbine power are presented in the section that follows, along with numerical simulations where it is identified and reproduced in the framework of UM. Acknowledging these biases, some efforts were made to characterize the effect of rainfall and wind velocity on turbine power. The final section concludes the study and summarizes the results.

## 2. Data and methods

### 2.1. Data and instrumentation

The Rainfall Wind Turbine or Turbulence project (RW-Turb, `https://hmco.enpc.fr/portfolio-archive/rw-turb/`), supported by Agence Nationale de la Recherche (ANR, French National research agency in English), is designed towards understanding the long and short-term effects of rainfall on wind power production, with simultaneous high-resolution measurements in an operational wind farm. Interested readers are directed to Gires et al. (2022) for an overview of the campaign. To briefly summarize, RW-Turb measurement campaign (at Pay d'Othe, 110 km southeast of Paris, France) consists of a meteorological mast (can be seen in Fig. 2b, at the right side) in a wind farm (jointly operated by Boralex: `https://www.boralex.com/our-projects-and-sites/` and JP Énergie Environnement: `https://pays-othe-89.parc-eolien-jpee.fr/`). Fig. 2a shows the location of the project; the nine wind turbines of the Pays d'Othe wind farm (aligned South-East of it and within a 4 km radius) are marked as black vertical crosses and the meteorological mast as a star (in the middle). Data from four Vestas V-90 (marked 1, 2, 8 and 9 in Fig.2a) are available, two are closer to the meteorological mast and two are farther from it ($\approx$ 3.5 km from mast). The five turbines of the Molinons wind farm in the North are also visible within the 5 km radius (grey vertical crosses). It should also be noted that a small grove is located just South of the mast at roughly 160 m; a larger one is on the East at roughly 100 m. Nearby the mast (i.e. within the 1 km radius), there is a small slope in the North-South direction. The meteorological mast consists of two sets of optical disdrometers (OTT Parsivel[2], 30 s; not used in current study), 3D sonic anemometers (ThiesCLIMA, 100 Hz), and mini meteorological station (1 Hz) at heights roughly 45 m and 80 m (managed by Hydrology Meteorology and Complexity laboratory of École des Ponts ParisTech - HM&Co, ENPC).

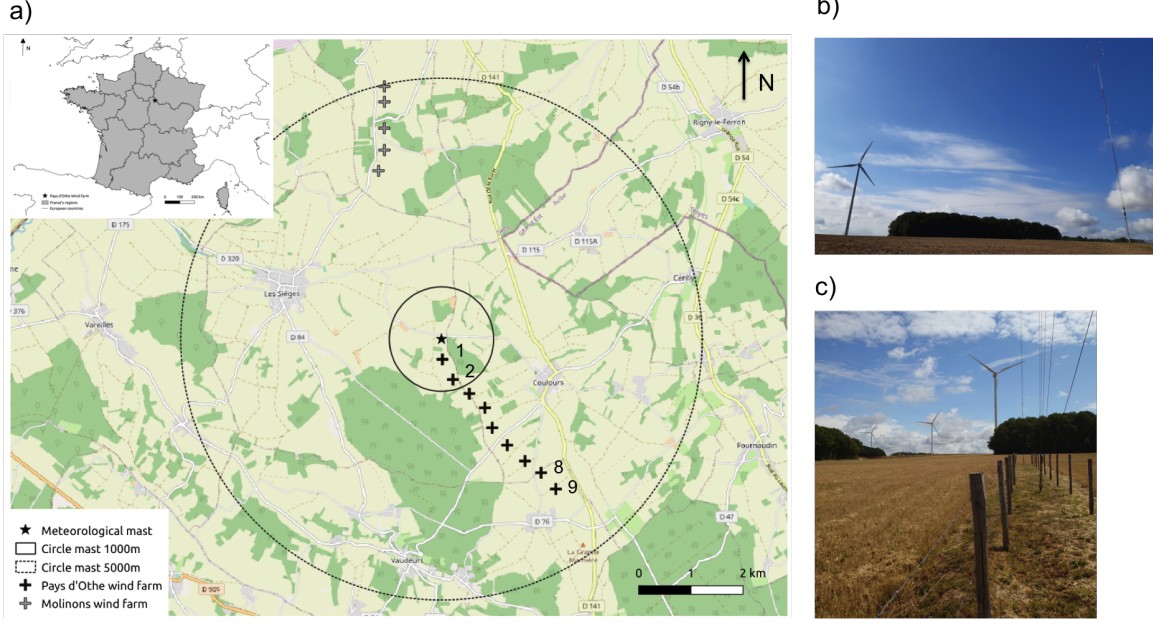

Figure 2: a) Map of the Pays d'Othe wind farm (inset: location in France), the meteorological mast is at the centre and turbines available are numbered - 1, 2, 8 and 9; b) Turbine 1 and the mast; c) Turbine 1 and 2 as seen from the bottom of the mast. Figures adapted from Gires et al. (2022).

Technical and working information of the turbine can be found in Vestas Wind Systems A/S (2023). The turbines have a rated power of $2.0\,\mathrm{MW}$ which is pitch regulated with variable speed. The hub height of the turbines is $80\,\mathrm{m}$, this is closer to the vertical height of upper set of devices on the mast ($\approx 78\,\mathrm{m}$). The turbines have a cut-in wind speed of $4\,\mathrm{m\,s^{-1}}$ and a rated wind speed of $12\,\mathrm{m\,s^{-1}}$. This can be see on power curves in Fig. 3 (last row) where the turbine register power at cut-in speed and maintain the rated power of $2000\,\mathrm{kW}$ after rated wind speed. The cut out speed of Turbine is at $25\,\mathrm{m\,s^{-1}}$ (the extreme x axis point of power curves); this is the speed at which turbine stops registering power. Generally the turbines register positive values of wind power, however, when the power retrieved from wind is less than that is required for working of turbine it registers negative power. These can be seen in the power curves as clusters around 0. Along with the wind power, turbine also provides information of local velocity (at a sampling measurement rate of $15\,\mathrm{s}$) which is used for internal regulation; this is used for plotting power curves in Fig. 3. The wind power data used for studies comes from four turbines by Boralex - 1 and 2 located closest to the mast (can be seen in Fig. 2b), and 8 and 9 located at the farthest end (at a sampling frequency of $15\,\mathrm{s}$).

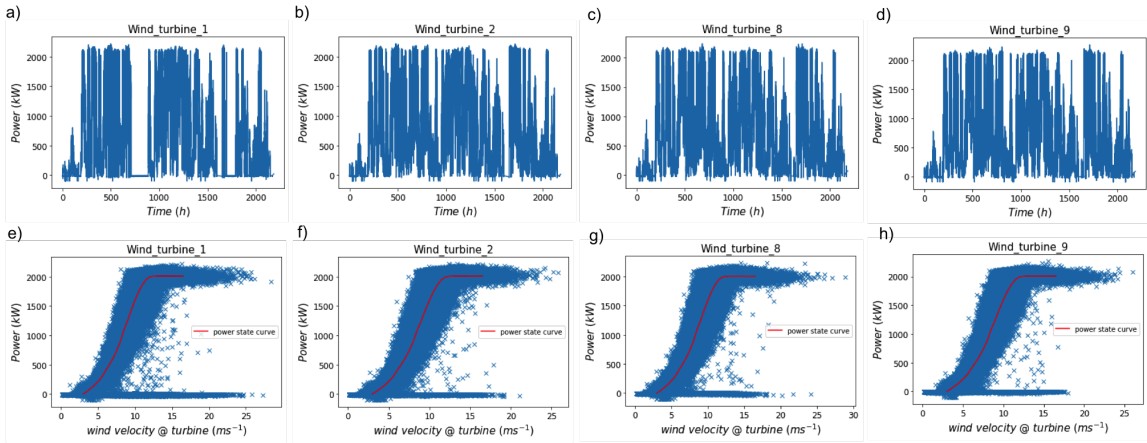

Figure 3: Time series of empirical turbine power (first row), and power vs velocity plot with theoretical state curve of turbine (second row) for the Wind Turbines (1, 2, 8, and 9) at Pays d'Othe.

The temporal evolution and power curves (power vs. velocity, expected curve provided by the manufacturer in red) for the turbines are shown in Fig. 3 for 3 months (from 01 Jan 2021 to 01 Mar 2021). There are instances where the turbine failed to produce any power and had to consume energy for its basic operation. This results in negative values in data, and for realistic analysis, they were considered as zero. This is why there are clustering of points at zero in the power curve (Fig. 3e - h).

In addition to the empirical power provided by turbine, the theoretical power available for extraction can be obtained by

$$P_a = \frac{1}{2}\rho A v^3 C_p \tag{1}$$

where $\rho$ is the air density at wind turbine height ($h_{hub}$), $A$ the swept area of turbine rotor, $v$ the wind velocity (ms$^{-1}$) approximated at turbine height and $C_p$ the power coefficient or Betz coefficient (for Vestas-90 examined here, $h_{hub}$ = 80 m; $A$ =6,362 m$^2$, and rated power is 2 MW; $C_p$ was taken as 0.593). Here, $P_a$ is estimated at the same sampling rate as that of $P_t$ (15 s) despite the 3D sonic anemometer and mini meteorological station registering data at finer sampling rates.

The value of air density is often approximated as 1.255 kgm$^{-3}$ (standard value at sea level, 15 °C). However, it is known to show fluctuations and reported to have an effect on power generation in varying levels (Jung and Schindler, 2019; Ulazia et al., 2018). For the purpose of this analysis, air density was considered as a varying quantity and estimated using the current official formula of the International Committee for Weights and Measures (CIPM), referred to as CIPM-2007 equation which accounts for humidity (Picard et al., 2008):

$$\rho(T,P,H_r) = \frac{PM_a}{Z(T,P,H_r)RT(K)} \left\{ 1 - x_v(T,P,H_r) \left[ 1 - \frac{M_v}{M_a} \right] \right\} \tag{2}$$

where $T$ (°C), $P$ (Pa) and $H_r$ ($0 \leq H_r \leq 1$) are temperature, pressure and humidity from Meteorological station at $h_{hub}$. Other derived parameters are

$T(K)$, air temperature (in K; from $T$)

$Z$, compressibility factor (a function of $T$ and $P$)

$R$, molar gas constant ($\mathrm{J\,mol^{-1}\,K^{-1}}$)

$x_v$, mole fraction of water vapour

$M_a$, molar mass of dry air ($\mathrm{g\,mol^{-1}}$)

$M_v$, molar mass of water ($\mathrm{g\,mol^{-1}}$)

### 2.2. Scaling analysis and UM framework

Spectral analysis is widely used for characterizing scaling properties; here, the second-order statistics of rain in the frequency domain were examined for power-law scaling as follows (Mandelbrot, 1982; Schertzer and Lovejoy, 1985).

$$E(k) \approx k^{-\beta} \tag{3}$$

where $k$ corresponds to the wave number and $\beta$ is the spectral exponent.

However, to fully characterize the complexity of the process, across its intensities and spatio-temporal variation, information on higher and lower-order statistics is required. For this, we use Universal Multifractals (UM), which relies on the assumption of the field being generated by an underlying cascade process with conserved statistical properties at each scale, while inheriting the scale invariant properties of Navier-Stokes equations (Schertzer and Lovejoy, 1987, 1989; Schertzer and Tchiguirinskaia, 2020). In this framework, the probability of a field exceeding a particular threshold across all scales is captured using the scale-invariant notion of singularity ($\gamma$) and for a multifractal field this probability scales according to the resolution ($\lambda$: the ratio of $L$, the outer scale, to $l$, the observational scale) with corresponding fractal codimension as the scaling exponent, $c(\gamma)$ (Schertzer and Lovejoy, 1987, 1988):

$$p\left(\varepsilon_\lambda \geq \lambda^\gamma\right) \approx \lambda^{-c(\gamma)} \tag{4}$$

This relation implies that statistical moments $q$ of the field also scale with resolution with moment scaling function $K(q)$ as (Schertzer and Lovejoy, 1987, 1988):

$$\langle \varepsilon_\lambda{}^q \rangle \approx \lambda^{K(q)} \tag{5}$$

Both function are related by Lengendre transform. For a conservative field in UM framework $K_c(q)$ can be fully determined with only two parameters, multi-fractality index $\alpha$ and mean intermittency codimension $C_1$ (Parisi et al., 1985).

$$K_c(q) = \begin{cases} \dfrac{C_1}{\alpha-1}(q^\alpha - q) & \alpha \neq 1 \\ C_1 q \ln q & \alpha = 1 \end{cases} \qquad (6)$$

$C_1$ measures clustering of average intensity across scales ($C_1 \in [0,1]$ for 1 dimensional fields); when $C_1 = 0$ the field is homogeneous with little variability. $\alpha$ measures how this clustering changes with respect to intensity levels ($\alpha \in [0,2]$); higher the value of $\alpha$, higher the variability, with $\alpha = 0$ being a monofractal field where intermittency of extreme is same as that of mean. If the UM parameters are known, co-dimension function of the conservative multifractal field, $c_c(\gamma)$ can also be obtained as using Lengendre transform (Schertzer and Lovejoy, 1987, 1988; Parisi et al., 1985):

$$c_c(\gamma) = \begin{cases} C_1 \left( \dfrac{\gamma}{C_1 \alpha'} + \dfrac{1}{\alpha} \right)^{\alpha'} & \alpha \neq 1 \\ C_1 \exp\left( \dfrac{\gamma}{C_1} - 1 \right) & \alpha = 1 \end{cases} \qquad (7)$$

where $\frac{1}{\alpha} + \frac{1}{\alpha'} = 1$.

For a non conservative field $\phi_\lambda$, i.e. a field whose average ($\langle \phi_\lambda \rangle$) changes with scales, a non-conservative parameter $H$ (not to be confused with the Hurst exponent; though for values > 0 both quantify long range correlations, the latter does not have a simple general expression for multifractal process, see appendix for more details) is used in the expression of scaling (Schertzer and Lovejoy, 1987, 1988; Lovejoy and Schertzer, 2013):

$$\phi_\lambda =^d \varepsilon_\lambda \lambda^{-H} \qquad (8)$$

where $=^d$ denotes equality in distribution: ($X =^d Y \Leftrightarrow \forall x : \Pr(X > x) = \Pr(Y > x)$) and $\varepsilon$ is a conservative field characterized with $C_1$ and $\alpha$. For a conservative field $H = 0$. For a non-conservative field with positive $H$, fractional differentiation is required to retrieve a coarser field. Similarly, from a non-conservative field with a negative value of $H$, the conservative field is retrieved through fractional integration. $H$ is related to the spectral slope $\beta$ (Eq. 3) via the relationship (Tessier et al., 1993):

$$\beta = 1 + 2H - K_c(2) \qquad (9)$$

The scaling behaviour of conservative multifractal fields can be examined using trace moment (TM) where log-log plot of upscaled fields against resolution $\lambda$ is taken for each moment $q$ (Eq. 5). The quality of scaling is given by the estimate $r^2$ of the linear regression; the value for $q = 1.5$ is used as reference. Double trace moment (DTM) is a more robust version of TM tailored for UM fields where the moment scaling function $K(q, \eta)$ of the

field $\varepsilon_\lambda^{(\eta)}$ (the initial field raised to power $\eta$ at maximum resolution and renormalized) is expressed as a function of multifractality index $\alpha$ (Lavallée et al., 1993).

$$\langle (\varepsilon_\lambda^{(\eta)})^q \rangle \approx \lambda^{K(q,\eta)} = \lambda^{\eta^\alpha K(q)} \tag{10}$$

From the above equation, value of $\alpha$ can be obtained as the slope of the linear part when $K(q,\eta)$ is represented for a given $q$ as a function of $\eta$ in log-log plot. Both TM and DTM techniques give reliable estimates as long as the $H < 0.5$ for the studied field.

Since multifractal processes are generated by cascade processes, the average values can get too concentrated over a certain area leading to spurious estimates of moments above a particular value of $q$ (at $q_D$, $q$ above which $K(q) \approx +\infty$). This effect is called divergence of moments. The convex nature of the functions $K(q)$ and $c(\gamma)$ are also limited by the sample size of data, or rather the maximum value of scale-invariant threshold or singularity ($\gamma_s$) and corresponding moment ($q_s$). Details on its computation can be found in (Schertzer and Lovejoy, 1992, 1989; Lovejoy and Schertzer, 2007). For reliable statistical estimates of the moment scaling function and hence the UM parameters, the moment orders used should not exceed $q_s$ (moment corresponding to maximum singularity) or $q_D$ (moment where divergence happens).

An in depth discussion of the methodological choices we have made, along with an overview of othermultifractal formalisms along with their strengths and weaknesses is provided in Appendix A.

## 3. Turbine power, biases and associated issues in data analysis

### 3.1. Turbine power and biases

For Vestas V-90, the rated power is 2 MW; this means that the maximum power the turbine can produce is 2000 kW. However, if we calculate the available power as per Eq. 1, there are many instances where it can go beyond the rated value (see Fig 4c). While analysing the variability of field using statistical methods, presence of instrumental limits in data (here an upper limit) can introduce biases. As briefly mentioned in introduction, an instrumental lower threshold in data can increase $\alpha$ and decrease $C_1$ (Jose et al., 2021). In addition to this, there is also the bias from presence of zeroes in data (Gires et al., 2014) (under estimation of $\alpha$, deterioration of scaling) which replaced negative values of turbine power. Fig. 4 shows the real and theoretical turbine power state curve along with the bias it poses in statistical analysis. Along with the power produced, turbine data also provides wind velocity at the hub (from a basic sensor installed on the hub), which is used for its internal monitoring. For research purposes, wind velocity from the 3D anemometer at the mast is more desirable as it offers a more reliable measurement (on almost same horizontal plane as turbine hub). Fig. 4a shows the empirical state curve of turbine with this velocity, and Fig. 4b shows the same curve with velocity from anemometer. There is considerably

more scatter with latter. It should be stated that turbines are not in the exact location of mast (Turbine 8 and 9 are $\approx 3.5$ km away) and hence approximation of wind velocity from mast (for computing theoretical power in Eq. 1) comes with some biases.

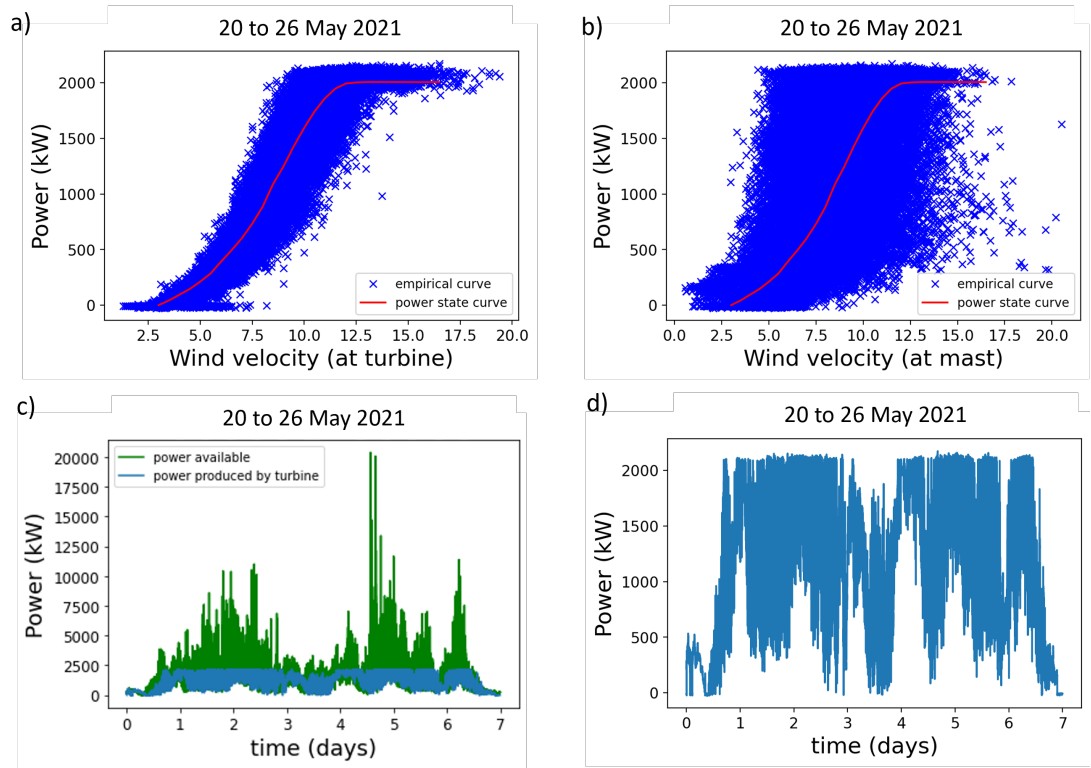

Figure 4: Illustration upper threshold (by virtue of rated power of 2000 kW) in power produced by turbine: a) Empirical and theoretical power state curve of turbine 1 with wind velocity from the turbine and b) Wind velocity from location 1 on the mast, c) Power produced by the turbine ($P_t$) and actual wind power available $P_a$, and d) Effect of rated power as threshold in time series and effect of negative values in $P_t$ for 1 week long data - 20 May 2021 to 26 May 2021.

From Fig. 4c and Fig. 4d, it can be clearly seen that the rated power imposes an upper threshold on turbine power ($P_t$) while power available ($P_a$) is the actual underlying field. For this week long series of $P_t$, 21.7 % of data was at upper threshold and 2.9 % were either zero or negative (taken as zeroes in analysis); this percentage was found to change according to data selected. Effect of these limits in UM analysis is shown in Fig. 5.a where the data in Fig. 4 is treated as an ensemble of 32 minutes. UM analysis was performed on direct fields as values of $H$ were within the acceptable limits ($H < 0.3$). A unique scaling regime from 15 s to 32 min was considered. Presence of rated power clips the values of field, and results in a reduced value of $\alpha$ for $P_t$ (Fig. 5a: $\alpha = 1.36$, $C_1 = 0.00715$) from that of $P_a$ (Fig. 5b: $\alpha = 1.93$, $C_1 = 0.01753$). Imposition of a similar threshold ($P_a <= 2000 = 2000$) on

$P_a$ was found to artificially reduce the estimates ($\alpha$ from 1.93 to 1.39, $C_1$ from 0.10753 to 0.0076) in Fig. 5c, bringing them closer to that of biased turbine power, $P_t$ (Fig. 5a). Even closer values of $\alpha$ were obtained when a lower threshold was also imposed (replacing $P_a$ values with zeroes at positions where $P_t$ was negative), giving $\alpha$ value of 1.35 and $C_1$ of 0.0077 (Fig. 5d). The results are compiled in Tab. 1. Scaling quality remained similar for all the cases mentioned here, with $r^2$ value (of TM curve at $q = 1.5$) remaining around 0.99 (second column of Fig. 5).

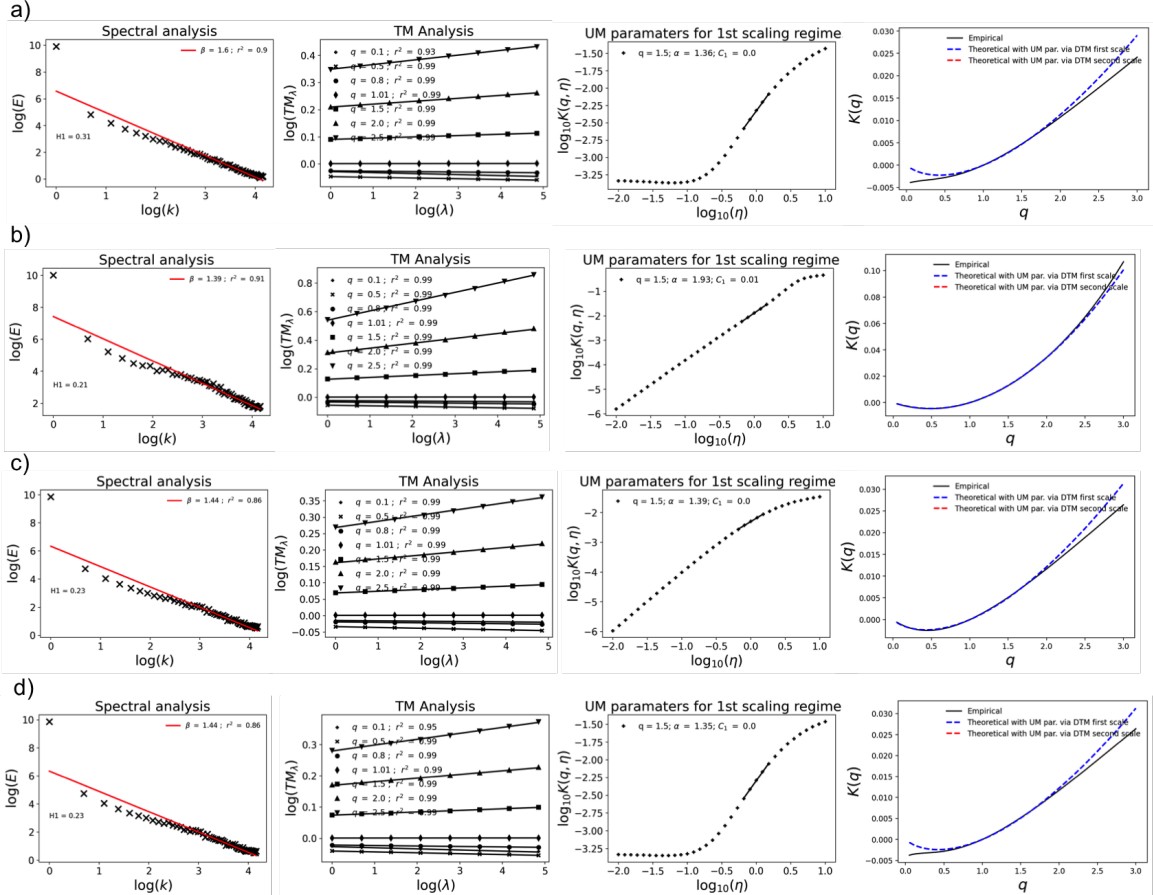

Figure 5: Spectral analysis (Eq. 3), TM analysis (Eq. 5), DTM curve (Eq. 10) and K(q) for a) power produced by turbine ($P_t$) which has intrinsic thresholds (upper: due to rated power, lower: due to negative values which are treated as zeroes), b) power available ($P_a$) which is the unbiased actual field, c) $P_a$ where an upper threshold is imposed at rated power of the turbine i.e. all values of $P_a$ above the rated power of turbine are artificially replaced by 2 MW (values > rated power = rated power), d) $P_a$ where an upper threshold and lower threshold (values set to zero where $P_t < 0$) are imposed based on the turbine values. Data used: time series from 20 May 2021 to 26 May 2021 with lowest time step of 15 s, with a sample size of 32 min.

| | | UM parameters | | | |
|---|---|---|---|---|---|
| **Field** | **threshold** | $\alpha_{DTM}$ | $C_{1,DTM}$ | $\beta$ | $H$ |
| $P_t$ | upper + lower | 1.36 | 0.00715 | 1.6 | 0.31 |
| $P_a$ | upper | 1.93 | 0.00753 | 1.39 | 0.21 |
| $P_a$ | lower | 1.35 | 0.0077 | 1.44 | 0.23 |
| $P_a$ | upper + lower | 1.35 | 0.0076 | 1.44 | 0.23 |

Table 1: Values of UM parameters for a) power produced by turbine ($P_t$) which has intrinsic thresholds (21.7 % at upper threshold, 2.9 % at lower threshold), b) power available ($P_a$) without any thresholds, c) $P_a$ with imposed upper threshold, d) $P_a$ with imposed upper threshold and lower threshold. Graphs can be seen in Fig. 4.

It should be noted that the effect of threshold could be different according to size of the sample and scaling regimes studied.In the same spirit as $\sigma_e$ in Jose et al. (2021), the effect of rated power as upper threshold in $P_t$ is explored here in the theoretical framework of UM. The effect on different scaling regimes as well as the additional complexity from the known effect of zeroes (Gires et al., 2012), though identified here, are not considered to avoid complexity.

### 3.2. Understanding the effect of upper threshold in UM framework

Let's take the upper threshold (rated power in this case) at the largest possible scale ratio as:

$$T = \Lambda^{\gamma_T} \tag{11}$$

where $\gamma_T$ is the singularity corresponding to threshold $T$, and $\Lambda$ the maximum resolution (length of time series). For multifractal fields, the probabilities of exceeding scale independent thresholds, $\lambda^\gamma$, scale with resolution, $\lambda$ (see Eq. 4). At the upper threshold $T$, this yields:

$$Pr(\varepsilon_\lambda \geq T) \approx \lambda^{-c(\gamma_T)} \tag{12}$$

If we set the upper threshold i.e. setting all the values of the field greater than $T$ equal to $T$ (represented here by this expression: $\{\varepsilon_\lambda \geq T\} = T$), the probability of having values greater than $T$, $Pr(\varepsilon_\lambda > T)$, becomes 0 reducing the above relation into $Pr(\varepsilon_\lambda = T) \approx \lambda^{-c(\gamma_T)}$. This leaves the value of $c(\gamma)$ equal to $+\infty$ for singularities above $\gamma_T$ (for $\gamma > \gamma_T$, $c(\gamma) = +\infty$). Here $c(\gamma_T)$ is the limiting non-zero value above which $c(\gamma)$ becomes $+\infty$.

This effect of upper threshold ($c(\gamma) \to +\infty$ for $\gamma > \gamma_T$) is similar to the effect of sampling dimension ($D_s$) in UM framework. The maximum observable singularity can be defined by taking probability at corresponding threshold as in Eq. 12:

$$Pr(\varepsilon_\lambda \geq \lambda^{\gamma_s}) \approx \frac{1}{N_s \lambda^D} \tag{13}$$

where $N_s$ is the number of samples and $\lambda^D$ is the number of values per sample. $N_s = \lambda^{D_s}$ ($D_s$ being the sampling dimension, which quantifies the number of independent samples with resolution $\lambda$; for one sample $D_s = 0$). Using the notions of $D_s$ and $D$, $\gamma$ corresponding to sampling resolution, $\gamma_s$ can be estimated from $c(\gamma_s)$, $c(\gamma_s) = D + D_s$ (Hubert and Carbonnel, 1989; Lovejoy and Schertzer, 2007).

The moment scaling function, $K(q)$ and codimension function, $c(\gamma)$ were discussed earlier in terms of UM parameters in Eq. 6 and Eq. 7. Both are equivalent functions and for multifractals, they are related by a simple Legendre transform (Parisi and Frisch, 1985; Schertzer and Lovejoy, 1993):

$$K(q) = \max_\gamma[q\gamma - c(\gamma)]$$
$$c(\gamma) = \max_q[q\gamma - K(q)] \tag{14}$$

Hence, for every singularity $\gamma$, there is a corresponding order of moment $q$ associated with it and vice versa: $q = c'(\gamma_q)$ & $\gamma = K'(q_\gamma)$.

When $\gamma > \gamma_s$, $c(\gamma) = +\infty$; by Legendre transform $K(q)$ becomes linear from $q > q_s = c'(\gamma_s)$

$$\gamma_s = \alpha' C_1 \left(\frac{D+D_s}{C_1}\right)^{\frac{1}{\alpha'}} - \frac{C_1}{\alpha - 1}$$
$$q_s = \left(\frac{D+D_s}{C_1}\right)^{\frac{1}{\alpha}} \tag{15}$$

In the case of sampling dimension, $c(\gamma)$ varies as follows

$$c(\gamma) = \left\{ \begin{array}{ccc} +\infty & for & \gamma > \gamma_s \\ D+D_s & for & \gamma = \gamma_s \\ c(\gamma) & for & \gamma < \gamma_s \end{array} \right\} \tag{16}$$

Similarly, at the presence of upper threshold here ($\{\varepsilon_\lambda \geq T\} = T$), $c(\gamma)$ reaches $+\infty$ at an earlier limiting value value $c(\gamma_T)$ where $\gamma_T < \gamma_s$ (Fig. 6a)

$$c_T(\gamma) = \left\{ \begin{array}{ccc} +\infty & for & \gamma > \gamma_T \\ c(\gamma_T) & for & \gamma = \gamma_T \\ c(\gamma) & for & \gamma < \gamma_T \end{array} \right\} \tag{17}$$

Here $\gamma_T$ is defined from the threshold as initially stated in Eq. 11, and $c_T(\gamma)$ is estimated as above. From this, the corresponding limit moment $_T$ can be obtained as in Eq.15.

$$q_T = \left( \frac{c(\gamma_T)}{C_1} \right)^{\frac{1}{\alpha}} \tag{18}$$

To summarize, in standard data analysis with sampling limitation, $c(\gamma)$ is bounded by a maximum value $c(\gamma_s)$ above which it becomes infinite. $K(q)$ which is connected to $c(\gamma)$ through Legendre transform (Eq. 14) becomes linear beyond this $q$ ($q \geq q_s$) value ($K(q) = (q - q_s)\gamma_s + K(q_s)$). Similarly, in this specific case, when an upper threshold is imposed ($\{\varepsilon_\lambda \geq T\} = T$), $K(q)$ becomes linear at an earlier value of $q$ (at $q_T < q_s$) defined by $\gamma_T$ (Fig. 6b).

$$K_T(q) = \left\{ \begin{array}{lll} \gamma_T(q - q_T) + K(q_T) & for & q > q_T \\ K(q_T) = q_T \gamma_T - c(\gamma_T) & for & q = q_T \\ K(q) & for & q < q_T \end{array} \right\} \tag{19}$$

In Double Trace Moment (DTM) technique, for a given $q$: $K(q, \eta) = K(q\eta) - qK(\eta)$, which for UM fields $= \eta^\alpha K(q)$. When no thresholds are applied $K(q, \eta)$ varies as

$$K(q, \eta) = \left\{ \begin{array}{lll} (q - 1)(D + D_s) & for & \eta \geq \eta_+(q) \\ \eta^\alpha K(q) & for & \eta < \eta_-(q) \end{array} \right\} \tag{20}$$

where $\eta_+(q)$ corresponds to the maximum values of $\eta$ above which $K(q, \eta)$ becomes a plateau due to sampling limitation (Eq. 15). To elaborate, $K(q, \eta)$ consists of two parts $K(q\eta)$ and $K(\eta)$, and $\eta_+(q)$ corresponds to the value of $\eta$ above which both are linear (which is $q_s$). The transition to the plateau starts at a lower value $\eta_-(q)$ (which is $q_s/q$) above which only $K(q\eta)$ is linear. In the presence of an upper threshold ($\{\varepsilon_\lambda \geq T\} = T$), DTM curve will be (Fig. 6c) :

$$K_T(q, \eta) = \left\{ \begin{array}{lll} (q - 1)c(\gamma_T) & for & \eta \geq \eta_+(q) \\ \eta^\alpha K(q) & for & \eta < \eta_-(q) \end{array} \right\} \tag{21}$$

where $\eta_+(q) = q_T$ and $\eta_-(q) = q_T/q$

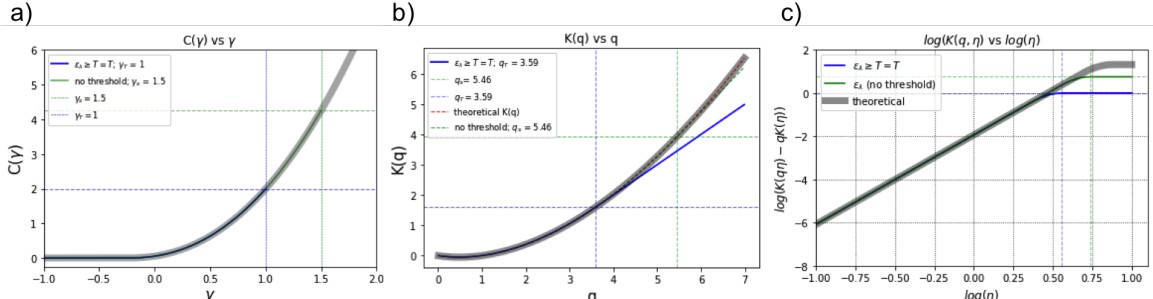

Figure 6: Influence of threshold on a) $c(\gamma)$ vs $\gamma$ curve: $c(\gamma)$ reaching $+\infty$ at $\gamma_T$ than $\gamma_s$, b) on DTM curve: $K(q)$ becoming linear at $q_T$ than $q_s$), and c) on $K(q)$ vs $q$ curve: $K(q,\eta)$ reaching upper plateau early. Arbitrary values were used for $\gamma_s$ and $\gamma_T$; UM parameter values of fields were taken as $\alpha = 1.8$ and $C_1 = 0.2$.

It is important to note here that the value of $K(q,\eta)$ doesn't reach the upper plateau abruptly at $q_T$ or $q_s$, rather, it flattens gradually starting from a value of $\eta = q_s/q$ or $q_T/q$ (as per value of $q\eta$ in $K(q,\eta)$). Presence of upper threshold shifts this starting point and decreases the range of possible values for estimation of $\alpha$ (slope of DTM curve), hence, presence of plateau will result in biased (reduced) estimates.

### 3.3. Numerical simulations

Underestimation in values of $\alpha$ due to application of upper threshold was already observed in Fig. 5c using real data. To understand this further, conservative multifractal fields ($H = 0$) were simulated using discrete cascades with values of UM parameters close to those observed for empirical power ($P_t$). Discrete cascades simulation here involves division of a parent structure into 'daughter' structures (retaining value of parent structure multiplied by a random factor, Chambers et al., 1976) iteratively following a non-infinitesimal scale ratio while maintaining the validity of Eq. 5 and Eq. 6.

For ease of contrast with simulations examining lower threshold in (Jose et al., 2021), values of $\alpha = 1.8$ and $C_1 = 0.2$ were used. UM analysis was implemented on ensembles (of sample size 128 and number of samples 100) by progressively applying the upper threshold till the percentages observed in $P_t$ ($\sim 30\%$). $K(q)$ becomes linear at earlier and earlier values of $q$ (after respective $q_T$) with increasing percentage of values at threshold can be seen in the third column (like in Fig. 6b). The DTM curve in second column shows that both $\alpha$ and $C_1$ are decreasing with progressive application of thresholds (from 0 to 30%, $\alpha$ decreased from 1.8 to 1.56 while $C_1$ decreased from 0.17 to 0.05).

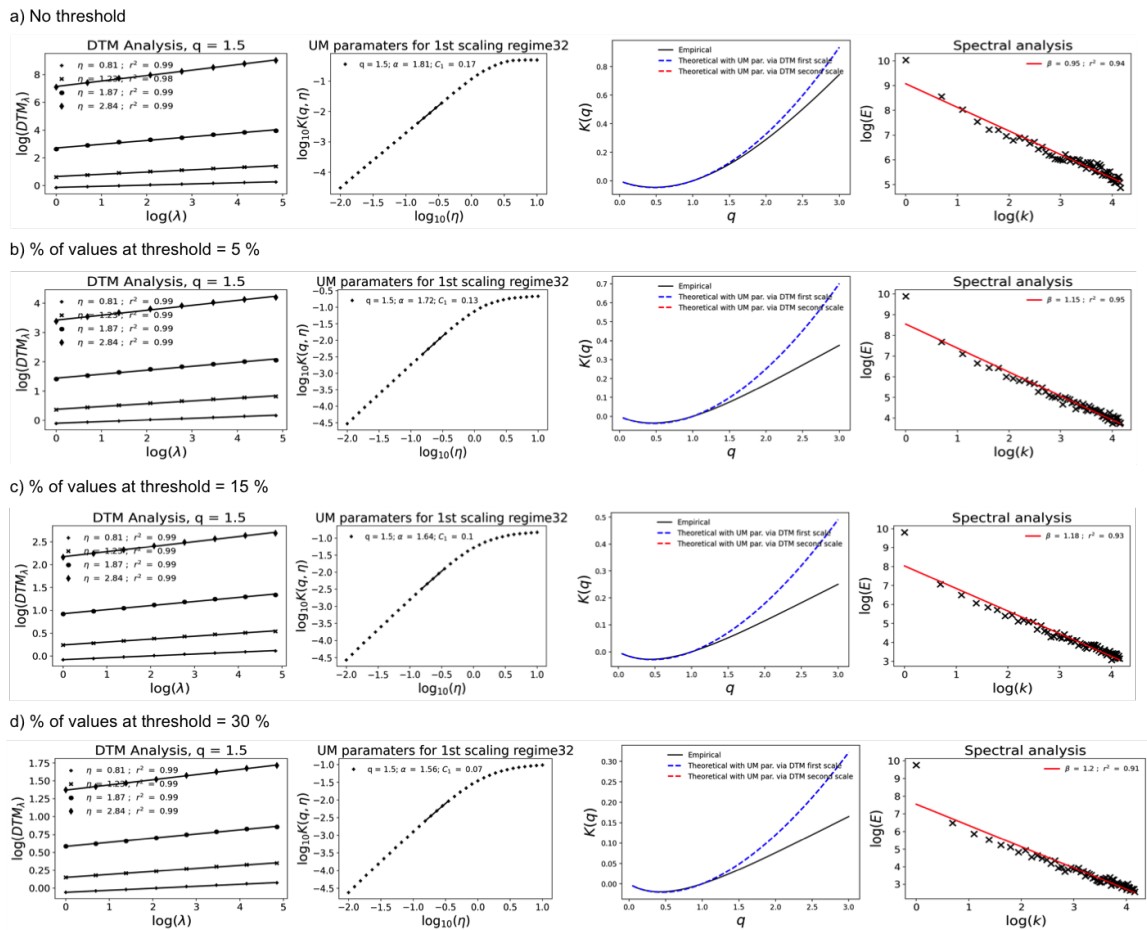

Figure 7: Effect of upper threshold illustrated using numerical simulations - discrete cascades of size 128 with 100 samples with $\alpha = 1.8$ and $C_1 = 0.2$ as input. Thresholds were applied progressively to the simulated field: a) no threshold, b) 5%, c) 15%, and d) 30%. Decrease in $\alpha$ and increase in $C_1$ with threshold can be seen from DTM curves (second column) of sub figures a to d.

| | UM parameters | | | |
|---|---|---|---|---|
| **% at threshold** | $\alpha_{DTM}$ | $C_{1,DTM}$ | $\beta$ | $H$ |
| 0% | 1.81 | 0.17 | 0.95 | 0.108 |
| 5% | 1.72 | 0.13 | 1.15 | 0.186 |
| 15% | 1.64 | 0.10 | 1.18 | 0.178 |
| 30% | 1.56 | 0.07 | 1.2 | 0.163 |

Table 2: Values of UM parameters for simulated fields with artificial imposition of upper thresholds.

While discussing this bias in the framework before, it was mentioned that the upper threshold was introduced at the maximum resolution (Eq. 11, $\Lambda^{\gamma_T}$). Since in practice, the lower scales in UM are obtained from averaging the outer scale (at maximum resolution), the threshold values (and hence $\gamma_T$) at each stage doesn't exactly correspond to the originally defined one. This is the reason for an increased 'transition part' (part of the curve from straight line to upper plateau) of the DTM curves (second column) in simulations here (more than that in Fig. 6c). When the estimation of $\alpha$ was forced at $\eta = 1$ (so that TM and DTM estimates are same) the bias in the values of $\alpha$ increased as the slope estimation moved to 'transition part'. For example, the already biased value of $\alpha$ at 30% threshold, 1.58 (slope at log $\eta$ between -0.1 and -0.5), got further reduced to 0.95 (slope around log $\eta = 0$). In this estimation, the $C_1$ remained moreover similar to previous estimates at all thresholds.

It is interesting to note that the trend here (only for $\alpha$) is not exactly the opposite of what was observed during numerical simulations with lower threshold in section (Jose et al., 2021). While the imposition of a lower threshold increased $\alpha$ and decreased $C_1$, the upper threshold here reduces both UM parameters. In the specific case of turbine power, $P_t$ (Fig. 5a), has a combination of upper threshold from rated power and lower threshold (zeroes) from negative power (the latter is not considered here). This, in practise, further reduces the range of available $\eta$ for estimation of $\alpha$ by imposing a lower plateau as well (see Fig. 5, third column). Also, the effect of this bias could be different when fluctuations of the fields are selected for retrieving conservative fields since the simulations were performed directly on conservative multifractal fields here. Since two consecutive power values can be the same, thanks to the rated power, taking fluctuations will yield zeroes in the field adding to the zero bias. The effect of both biases could be different when aggregate power of the wind farm is considered as well, this is also not explored here.

*3.4. Data analysis reducing the biases*

So far, the effect of thresholds in UM analysis has been illustrated in the framework of UM. To have a better idea of its effect on scaling, UM analysis was performed on a longer

series of power. For this, a 3-month long continuous series was taken (from 01 Jan 2021 to 01 April 2021) and UM analysis was performed on $P_a$ and $P_t$ as ensembles of sample size 128 (32 min). Fig. 8a and Fig. 8b shows the curves for $P_a$ ($\alpha$: 1.93; $C_1$: 0.01) and $P_t$ ($\alpha$: 1.11 ; $C_1$: 0.0042) respectively. Considering $P_a$ as the underlying field, the effects of thresholds (upper and due to zeroes) in $P_t$ can be seen in the DTM curve (Fig. 8b, second row). The lower plateau corresponds to presence of negative power in data (which were replaced by zeroes) and the low value of $\alpha$ is due to $\eta$ being in the transition range (as already seen in Fig. 6c). Fig. 8c and Fig. 8d shows the UM analysis for the same data but by removing the columns having thresholds. In $P_t$ ensemble data, the columns with thresholds, zeroes and repetition of data were removed by using a limit of 0.01%. For example, columns with more than 0.01% values $\geq 1600$ were removed to be on the safe side of analyzing data without the influence of threshold. For a more accurate comparison, the same columns were removed from $P_a$ as well; the results are shown in Fig. 8c ($\alpha$: 1.76; $C_1$: 0.0095) and Fig. 8d. It can be seen that the lower plateau has disappeared for $P_t$ and that the value of UM parameters ($\alpha$: 1.5749 from 1.11; $C_1$: 0.00554 from 0.0042) has improved; this also increased the value of $\beta$ (1.8 from 1.6), and consequently that of $H$ (0.41 from 0.3). It should be noted that the values of UM parameters get closer to that of $P_a$ but are not the exact values. This suggests that there are slight differences in the properties of both fields, even though they appear comparable when biases are removed.

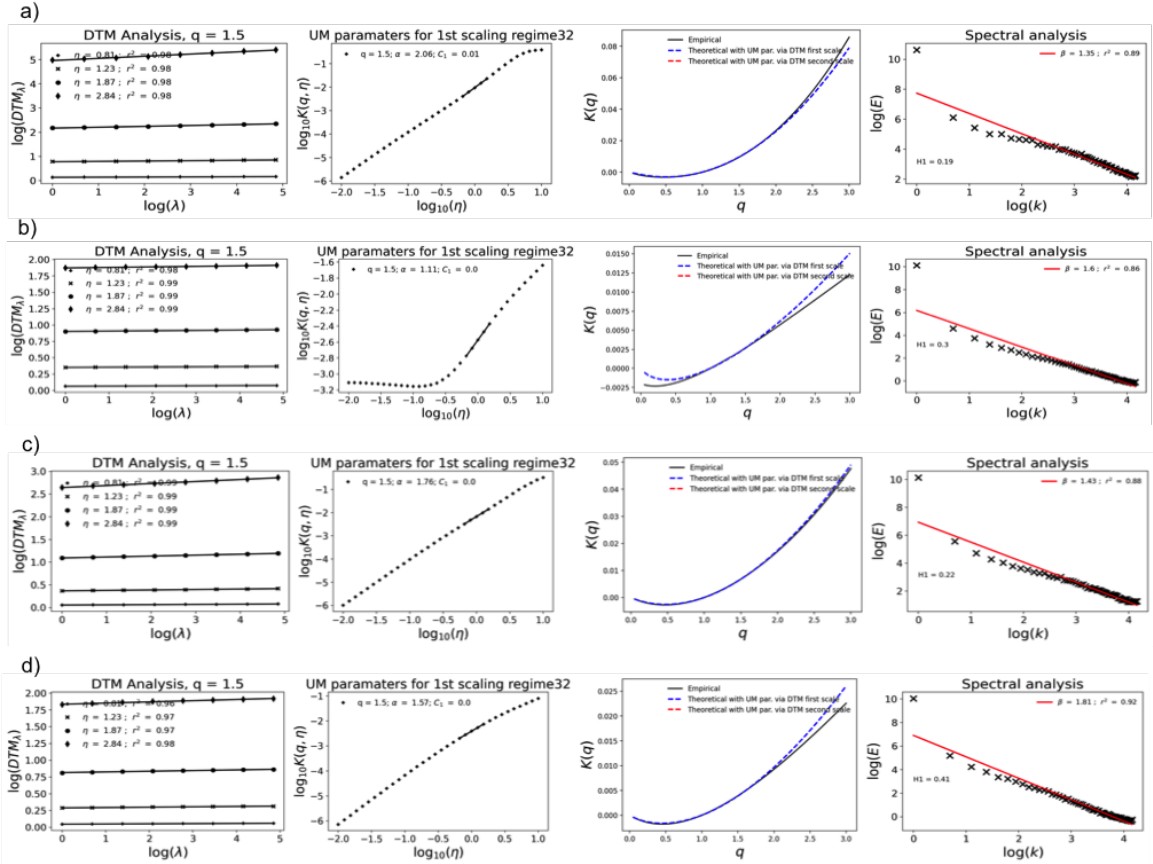

Figure 8: UM analysis of data from 01 Jan 2021 to 01 April 2021 as ensembles of sample size 128 (32 min): a) $P_a$ (as the data is), b) $P_t$ (as the data is), c) $P_a$ (columns with threshold removed on the basis of $P_t$ data below), and d) $P_t$ with columns of upper threshold and zero data removed.

## 4. Conclusion

Wind turbines are designed to work at a rated power for optimal production of power as well as their safe functioning. This inadvertently creates an upper threshold in output data of power production and such a effect induces biases in statistical analysis, especially
when the small scale non linear variability and intermittency are to be studied. Backdrop of this study followed the campaign in Gires et al. (2022) where the main objective of was to analyze turbine power, $P_t$, as a temporal field and to gain insights into its correlation with rainfall, which is poorly understood, and also with other meteorological fields, across scales, using the data averaged to these reliable frequencies. However, the direct analysis
of empirical turbine power (using the framework of UM) was found to be difficult since the output from wind turbines are limited by a maximum or rated power. In time series analysis

this acts as an upper threshold resulting in reduced estimates of UM parameters than those of theoretically available wind power ($P_a$) for extraction. The reason for this decrease was identified within the framework of UM, and is illustrated using theoretical formulations. The same has been confirmed through simulations of conservative multifractal fields, as well. Basically, the presence of an upper threshold introduces an upper plateau in DTM curve, similar to the one due to the sampling dimension, but it begins at a lower value of $\eta$. This reduces the range of available $\eta$ for estimation of the slope and hence results in a biased value of $\alpha$ (reduced $\alpha$ and $C_1$). Also, when it comes to the empirical power produced by turbines, the biases are twofold since a lower threshold (albeit to very less extend) is also involved since the turbine does not necessarily always produce power and has moments that involve only operational consumption (leaving power production values negative). Since, UM in its usual form is not designed to handle negative values, based on how these values are managed (taken as zero here), the values of $\alpha$ will be further biased due to the effect of lower threshold.

Though these biases are identified, as of now, no solutions are available to account for them; and more methodological developments are required for this solution. Same is required for understanding the combined effect of both biases. It is also worth mentioning that such an upper threshold is very likely to affect other statistical analysis relying on scale invariance as well. However, this is beyond the scope of current paper and and would require separate investigations. Due to the presence of above said biases in $P_t$, the actual wind power available at the turbine hub for extraction ($P_a = f(v, \rho)$, Eq. 1) was used as a proxy to understand the small scale variability in follow up UM analysis. Since, the presence of thresholds in data - imposed by limitations of operations as well as measurement - exists in many geophysical situations, understanding them is important in retrieving the actual characteristic parameters as well as modelling them. In the case explored here, since the characterization of power production is already suffering from various influences that are poorly understood and accounted for, understanding the biases in data treatment will help avoid more uncertainties.

## Competing interests

At least one of the (co-)authors is a member of the editorial board of Nonlinear Processes in Geophysics.

## Acknowledgement

The authors greatly acknowledge partial financial support from the Chair of Hydrology for Resilient Cities (endowed by Veolia) of the École nationale des ponts et chaussées, and the ANR JCJC RW-Turb project (ANR-19-CE05-0022-01).

## Appendix A: Diversity of multifractal formalisms and theoretical choices

In response to questions from referees, and therefore potential questions from readers, we felt it necessary to discuss in greater depth the methodological choices we have made. The purpose of this appendix is therefore to provide, in a fairly autonomous way to ease the reading, a better overview of the formalisms, highlighting their strengths and weaknesses, their common features and their diversity. This appendix is also valid for part 2 of this paper.

### A.1. Deterministic/stochastic multifractals and dimension/codimension formalisms

In this paper and its companion paper (Jose et al., 2024a,b) we use a stochastic multifractal framework (Schertzer and Lovejoy, 1984b,a, 1989, 1992), for the fundamental reason it is much more general than a deterministic multifractal framework (Parisi et al., 1985; Halsey et al., 1986). This strong difference is fundamentally due to the number of samples required to get reliable information. While a unique sample is sufficient for a pure deterministic process, the determination of extremes of a stochastic process may require a very large number of samples. This difference is also illustrated by the fact that a stochastic event has a finite occurence frequency, while it may occur on an infinite number of samples. This results from the fact that a probability frequency can be understood as the (finite) limit of the ratio of two diverging numerations: 'favorable cases' vs. 'all cases'. If both numerations are scaling with a dimension exponent, then the frequency scales with the difference of these dimensions, usually called co-dimension. This can be written down as follows for a process $\varepsilon_\lambda$ at resolution $\lambda = L/\ell$ (outer scale $L$, observation scale $\ell$ ) when assessing its probability to diverge faster than $\lambda^\gamma$, i.e. has a singularity $\gamma$:

$$Pr(\varepsilon_\lambda \geq \lambda^\gamma) \approx \frac{N_\lambda(\varepsilon_\lambda \geq \lambda^\gamma)}{N_\lambda} \approx \frac{\lambda^{D(\gamma)}}{\lambda^D} = \lambda^{D(\gamma)-D} = \lambda^{-c(\gamma)}, \tag{A.1}$$

where the codimension $c(\gamma)$ and the dimension $D(\gamma)$ satisfy:

$$D = c(\gamma) + D(\gamma) \tag{A.2}$$

where $D$ is the embedding dimension of the process and thus generalise the definition of the codimension $C(A)$ of a $D(A)$-dimensional subspace $A$ in a $D$-dimensional vector space:

$$D = D(A) + C(A) \tag{A.3}$$

Equation A.1 provides a first insight on the fundamental fact that scaling exponents of probabilities are codimensions, while those of numerations are dimensions.

It seems paradoxical that the dimension/deterministic multifractal formalism was introduced to explain the observed nonlinearity of statistical scaling exponents, precisely that of

the velocity structure functions (Anselmet et al., 1984), which are the *statistical* moments of the velocity increments. This was done with the help of strong assumptions: the singularities of the velocity increments, defined as local Holder exponents, were assumed to be geometrically and rather deterministically distributed over embedded fractals. In many respects, the $f(\alpha)$ formalism (Halsey et al., 1986), which deals with multifractal strange attractors, further emphasised this implicit nonrandom and geometric framework. A dimension formalism such as $f(\alpha)$ formalism is formally related to codimensions $(\gamma, c(\gamma))$ according to:

$$\alpha_D = D - \gamma; f_D(\alpha_D) = D - c(\gamma) \qquad (A.4)$$

The subindex $D$ is introduced to $\alpha$ and $f$ for two reasons:

- both $\alpha$ and $f$ depend on the embedding dimension $D$, e.g. by taking cuts of dimensions smaller than $D$, while $(\gamma, c(\gamma))$ do not depend on it

- another 'historical' $\alpha$ that has a quite different meaning, as shortly recalled below.

The same dependence on $D$ occurs for the scaling exponent $\tau(q)$ of the partition function (Hentschel and Procaccia, 1983; Jiang et al., 2019), which is strongly related to that of the statistical moment scaling exponent $K(q)$:

$$< \varepsilon_\lambda > \approx \lambda^{K(q)} \qquad (A.5)$$

as follows:

$$\tau_D(q) = (q-1)D - K(q) \qquad (A.6)$$

*A.2. Partial equivalence between dimension and codimension formalisms*

In fact we implicitly used the partial equivalence between both formalisms for the introduction of the codimension (Eq.A.1). Before insisting on its partiality, let us stress it is merely defined by Eqs.A.4, a very broad but rather straightforward generalisation of Eq.A.3. The equations A.4 thus define the framework transformation to go from a formalism to the other one.

Unfortunately, this equivalence is only partial because the dimension framework is much more limited than the codimension one. A major difference is that the numeration dimension $D(\gamma)$ is both bounded below and above ($0 \leq D(\gamma) \leq D$), while the codimension $c(\gamma)$) being statistically defined as the opposite of the scaling exponent of the probability exponent has in general no upper bound ($0 \leq c(\gamma)$. For instance, $c(\gamma)) = \infty$ merely corresponds to a singularity $\gamma$ that almost never happens, including at finite resolution. According to Eq.A.4, as soon as $c(\gamma) > D$ would correspond to a negative dimension $D(\gamma)$.

 *A.3. Universality and stochasticity*

One important feature of the stochastic framework is that it provides universal behaviour, i.e. processes that are both attractive and stable, as well as determined by only a limited number of parameters. We briefly recall that this is the case of the 'Universal Multifractals' (UM; Schertzer and Lovejoy (1987, 1997)) that satisfy a broad generalisation of the central limit theorem: their generators are attractive and stable through renormalised summations. We recall that their statistics are defined by the three parameters that follow and which are physically meaningful:

- the scaling exponent $H$ [1] of the mean field. When $H = 0$, the mean field is strictly scale invariant and the field is said conservative. $H \neq 0$ often results from a fractional integration of this order of a conservative field;

- the codimension $C_1 \geq 0$ of the mean field. It measures the mean intermittency, i.e. how the mean fluctuations are increasingly concentrated scale by scale. When $C_1 = 0$ there is no intermittency and the field is statistically homogeneous;

- the multifractal index $0 \leq \alpha \leq 2$. It measures the variability of the intermittency when departing from the mean field. When $\alpha = 0$ the field is uni/mono- fractal, $\alpha = 2$ corresponds to a maximal intermittency and to the so-called lognormal model. $\alpha$ is also the Levy stability index (Lévy, 1937) of the cascade generator.

The corresponding UM scaling moment function is therefore:

$$K q) = -qH + K_c(q); K_c(q) = C_1 \frac{q^\alpha - q}{\alpha - 1} \tag{A.7}$$

where $K_c(q)$ denotes the scaling moment function of a conservative UM field. We consider that the case studies of the text body confirm that Eq.A.7 allows a much richer data analysis than the deterministic indicators frequently used, such as $\Delta \alpha_D = \alpha_{D,max} - \alpha_{D,min}$.

*A.4. Hurst exponent and its multifractal generalisations*

Scaling time series analysis have been strongly focused on the estimation of the historical Hurst exponent $H$ (Hurst, 1951) in particular with respect to its critical value $H = 0.5$ supposed to discriminate long range dependency and persistence ($H > 1/2$) from short one and anti-persistence ($H < 1/2$). The multifractal ideology has ruined the dogma of its uniqueness and justified that divergent estimates where not accidental, but resulted from a given physics, that of intermittency. Among many ways to define a Generalised Hurst

---

[1]As discussed below it is related to the historical Hurst exponent, although being rarely identical to it, see Eq.A.9 and associated comments.

Exponent (GHE, Gómez-Águila and Sánchez-Granero (2021)) it is straightforward to consider the scaling exponent $H(q)$ of the $q^{-th}$ root of the (absolute) $q^{-th}$ order statistical moment of the field [2]:

$$H(q) = -K(q)/q \tag{A.8}$$

This definition is very generic and is an effective measure of the evolution of the $q^{-th}$ order statistical moment, if any, with respect to its order $q$. In particular, the uniqueness of $H$ ($H(q) \equiv H$) is recovered for fractional integrations of homogeneous processes. On the contrary, fractional integrations of multi-fractal processes yield a non-constant part, e.g. for a fractional integration of a conservative field with the scaling moment function $K_c(q)$:

$$H(q) = H - K_c(q)/q \tag{A.9}$$

In the generic case of universal multifractals, $K_c(q)$ depends only on the universal parameters $C_1$ and $\alpha$ (Eq.A.7). Because $H(2)$ is often considered as the historical Hurst exponent (Kantelhardt, 2002), it is equal to $H$ only for homogeneous processes according to Eq.A.9.

The main drawback of the generalised Hurst exponent $H(q)$ is that it gives access to the statistics of the cascade generator less directly than with the scaling moment function $K(q)$. This may explain why many GHE studies have limited outputs due to a lack of theoretical guidance, e.g. by only providing raw statistics of $H(q)$ such as its minimum and maximum over a given range of $q$.

### A.5. Detrending frameworks

Multifractal Detrended Fluctuation Analysis (MFDFA), (Kantelhardt, 2002) is a popular scaling analysis technique that explicitly uses the concept of Generalised Hurst Exponent. However, this is not directly done on the field of interest, but on the standard deviations of the residues of polynomial regressions on the running sum of the fluctuations of the original time series. It thus corresponds to a multifractal extension of the (fractal) Detrended Fluctuation Analysis (DFA, Peng et al. (1994)), as it is generally presented.

Let us provide some details about this. Let $Y(i)$ be the cumulative fluctuation of the original time series $x(k)$ of mean value $<x>$ :

$$Y(i) = \sum_{k=1}^{i} [x(k) - <x>] \tag{A.10}$$

The series is split into $N_s$ non-overlapping sub-series of finite size $s$ and a detrending polynomial $y_v(i)$, with constant order $m$, is fitted in each sub-series by least squares. This yields

---

[2]The minus sign that appears in this relation is only due to the lhs is a scaling exponent with respect to scales, whereas the rhs is in respect to resolution.

the following root mean square variation over the $v^{th}$ sub-series:

$$F(v,s) = \left[\frac{1}{s}\sum_{i=1}^{s}\{Y[(v-1)s+i]-y_v(i)\}^2\right]^{1/2} \tag{A.11}$$

and averaging over the $N_s$ sub-series of size $s$ yields the total variation for the DFA analysis:

$$F(s) = \left[\frac{1}{N_s}\sum_{v=1}^{N_s}F^2(v,s)\right]^{1/2} \tag{A.12}$$

The generalisation to MFDA is straightforwardly obtained by introducing the statistical order $q$ instead of 2:

$$F_q(s) = \left[\frac{1}{N_s}\sum_{n=1}^{N_s}[F^2(v,s)]^{q/2}\right]^{1/q} \tag{A.13}$$

and the estimate of $H(q)$ is obtained by the logarithmic slope of $F_q(s)$:

$$F_q(s) \approx s^{H(q)} \Leftrightarrow H(q) \approx \frac{\ln F_q(s)}{\ln s} \tag{A.14}$$

It is similar, but not identical to the scaling of the trace-moment of the original field.

### A.6. Defering to future work

A priori, $H(q)$ is not unique, since it may depend like $F_q(s)$ on the order $m$ of the detrending polynomials $y_v(i)$ and there is no obvious theoretical guidance of how to choose
this order. In addition, the cumulative fluctuation obviously increases the order of integration $H$ by a unit (Eq.A.10). Conversely, the obtained estimates (Eq.A.14) must be reduced by the same amount to indirectly estimate a $H(q)$ for the original series. The most serious theoretical drawback is the linear decomposition into local polynomial trends and stochastic fluctuations. Moreover, as the former is maximised by least squares, the importance of
the fluctuations is minimised even though they are initially at the the centre of the analysis. Because of these many issues we defer the MFDA analysis and variants of our data to future work.

### References: Appendix A

Anselmet, F., Gagne, Y., Hopfinger, E., Antonia, R., 1984. High-order velocity structure functions
in turbulent shear flows. Journal of Fluid Mechanics 140, 63–89.

Gómez-Águila, A., Sánchez-Granero, M., 2021. A theoretical framework for the tta algorithm. Physica A: Statistical Mechanics and its Applications 582, 126288.

Halsey, T.C., Jensen, M.H., Kadanoff, L.P., Procaccia, I., Shraiman, B.I., 1986. Fractal measures and their singularities: The characterization of strange sets. Physical review A 33, 1141.

Hentschel, H.G.E., Procaccia, I., 1983. The infinite number of generalized dimensions of fractals and strange attractors. Physica D: Nonlinear Phenomena 8, 435–444.

Hurst, H.E., 1951. Long-term storage capacity of reservoirs. Transactions of the American society of civil engineers 116, 770–799.

Jiang, Z.Q., Xie, W.J., Zhou, W.X., Sornette, D., 2019. Multifractal analysis of financial markets: A review. Reports on Progress in Physics 82, 125901.

Jose, J., Gires, A., Roustan, Y., Schnorenberger, E., Tchiguirinskaia, I., Schertzer, D., 2024a. Part 1: Multifractal analysis of wind turbine power and the associated biases. Nonlinear Processes in Geophysics Discussions 2024, 1–24. URL: `https://npg.copernicus.org/preprints/npg-2024-5/`, doi:`10.5194/npg-2024-5`.

Jose, J., Gires, A., Schnorenberger, E., Roustan, Y., Schertzer, D., Tchiguirinskaia, I., 2024b. Part 2: Joint multifractal analysis of available wind power and rain intensity from an operational wind farm. Nonlinear Processes in Geophysics Discussions 2024, 1–36. URL: `https://npg.copernicus.org/preprints/npg-2024-6/`, doi:`10.5194/npg-2024-6`.

Kantelhardt, J., 2002. Multifractal deterended fluctuation analysis of nonstationary time series. Physica 316, 81–91.

Lévy, P., 1937. Théorie de l'addition des variables aléatoires. Collection des monographies des probabilités, Gauthier-Villars. URL: `https://books.google.fr/books?id=1d6iuAEACAAJ`.

Parisi, G., Frisch, U., et al., 1985. A multifractal model of intermittency. Turbulence and predictability in geophysical fluid dynamics and climate dynamics , 84–88.

Peng, C.K., Buldyrev, S.V., Havlin, S., Simons, M., Stanley, H.E., Goldberger, A.L., 1994. Mosaic organization of dna nucleotides. Physical review e 49, 1685.

Schertzer, D., Lovejoy, S., 1984a. Elliptical turbulence in the atmosphere, in: Symposium on Turbulent Shear Flows, 4 th, Karlsruhe, West Germany, p. 11.

Schertzer, D., Lovejoy, S., 1984b. On the dimension of atmospheric motions. volume 505. Elsevier, North-Holland, Amsterdam.

Schertzer, D., Lovejoy, S., 1987. Physical modeling and analysis of rain and clouds by anisotropic scaling multiplicative processes. Journal of Geophysical Research: Atmospheres 92, 9693–9714. URL: `https://agupubs.onlinelibrary.wiley.com/doi/abs/10.1029/JD092iD08p09693`, doi:`10.1029/JD092iD08p09693`.

Schertzer, D., Lovejoy, S., 1989. Nonlinear Variability in Geophysics: Multifractal Simulations and Analysis. Springer US, Boston, MA. pp. 49–79. URL: `https://doi.org/10.1007/978-1-4899-3499-4_3`, doi:10.1007/978-1-4899-3499-4_3.

Schertzer, D., Lovejoy, S., 1992. Hard and soft multifractal processes. Physica A: Statistical Mechanics and its Applications 185, 187–194. URL: `https://www.sciencedirect.com/science/article/pii/037843719290455Y`, doi:https://doi.org/10.1016/0378-4371(92)90455-Y.

Schertzer, D., Lovejoy, S., 1997. Universal multifractals do exist!: Comments on âa statistical analysis of mesoscale rainfall as a random cascadeâ. Journal of Applied Meteorology 36, 1296 – 1303. URL: `https://journals.ametsoc.org/view/journals/apme/36/9/1520-0450_1997_036_1296_umdeco_2.0.co_2.xml`, doi:10.1175/1520-0450(1997)036<1296:UMDECO>2.0.CO;2.

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
