# Peer review of "Part 1: Multifractal analysis of wind turbine power and the associated biases"

_Nonlinear Processes in Geophysics, 2024_

## Referee Comment (RC1)

The authors characterize the small-scale fluctuations in wind power production using data from an operational wind farm at 70 Pays d'Othe, 110 km southeast of Paris, France, and Universal Multifractals framework. The main objective of this article is to highlight some biases found during multifractal analysis and their influence on the direct statistical analysis of turbine power. Then, with numerical simulations and analytical expressions based on the UM framework, the authors support the influence of a higher threshold in the power available on the biases found.

**Major issues**

1. The summary of the article is too long, around 33 lines. The authors should be more concise in the summary because several of their ideas would be better in the introduction.
2. Eq. 8 presents the multifractal behavior for a non-conservative field with parameter H. Also, it is known that other important measures in multifractality are the Renyi entropy or the generalized fractal dimension (see https://doi.org/10.1088/1361-6633/ab42fb). Therefore, there remain two important points to be addressed in this direction and that should be mentioned in the article to establish future work directions of this article: A) What is the relationship of the parameter H with other multifractal measures such as the Hurst exponent generalized or the generalized fractal dimension? B) What relationship exists between the trace moment (TM) or double trace moment (DTM) method with other methods with which multifractal exponents are estimated, for example multifractal detrended fluctuation analysis (MF-DFA), total triangle areas (TTA), generalized Hurst exponent (GHE), among others (see for a review https://doi.org/10.1016/j.physa.2021.126288)?
3. Section 3.2 establishes a way to understand the effect of a higher threshold in the Universal Multifractals framework. The authors could highlight the difference of this method with others where the bias introduced in the estimation of multifractal exponents is considered (see for example https://doi.org/10.1016/S0378-4371(96)00165-3 and https://doi.org/10.1103/PhysRevE.95.042311).

**Minor issues**

1. Eq. (4), (5), (6), (7) and (8), do not have explicit references from which they were taken before being placed as was done for Eq. (3). The above, although it is a minor change, is suggested so that those readers who do not know much about the Universal Multifractals approach can inquire about it, and therefore, for the article to have a greater scope.

2. In equation (7) the restriction of $1/\alpha + 1/\alpha^{'}=1$ is not clear, it is suggested to clarify this restriction and what $\alpha^{'}$ represents in said equation.

3. In the title of Figure 5, panel (c), it is not clear the upper threshold condition placed on the power available, the authors could revise this part or clarify this condition in words (*"values>rated power = rated power"*).

4. Section 3.2, which explains the effect of a higher threshold in the Universal Multifractals framework, should go as one more subsection of section 2. The above, to give greater cohesion to the article since at this time the manuscript remains disconnected between the results obtained with the empirical data of the wind farm and the numerical simulations.

Furthermore, it seems to me to be a new part of the article and one that deserves to be highlighted.

5.  In section 3.3 it is not entirely clear how each simulation of the discrete cascades was generated. Thus, it is necessary to establish in greater detail how each of these ensemble simulations were achieved.

---

## Author Comment (AC1)

**Response to comments from editors and reviewers**

Part 1: Multifractal analysis of wind turbine power and the associated biases (npg-2024-5)

preprint in Nonlinear Processes in Geophysics from 02 Feb 2024

Editor's decision received on
* * *
The authors would like to thank the reviewer for evaluating the paper and providing a detailed feedback. We appreciate the positive feedback and have tried our best to incorporate them in the updated manuscript. Hopefully the modifications are in accordance with the reviewer's expectations and quality of the journal.

Please find below our point-by-point response to the comments (Reviewer comments are shown in black and author responses are in blue).

**Anonymous Referee #1, 11 Mar 2024**

The authors characterize the small-scale fluctuations in wind power production using data from an operational wind farm at 70 Pays d'Othe, 110 km southeast of Paris, France, and Universal Multifractals framework. The main objective of this article is to highlight some biases found during multifractal analysis and their influence on the direct statistical analysis of turbine power. Then, with numerical simulations and analytical expressions based on the UM framework, the authors support the influence of a higher threshold in the power available on the biases found.

Thank you very much for taking the time to read and review our manuscript. We greatly appreciate the feedback. Please find our response to the points raised below (in blue).

*Major issues*

1. The summary of the article is too long, around 33 lines. The authors should be more concise in the summary because several of their ideas would be better in the introduction.

    Thank you. The abstract is now made concise in 17 lines for better readability.

2. Eq. 8 presents the multifractal behavior for a non-conservative field with parameter H. Also, it is known that other important measures in multifractality are the Renyi entropy or the generalized fractal dimension (see `https://doi.org/10.1088/1361-6633/ab42fb`). Therefore, there remain two important points to be addressed in this direction and that should be mentioned in the article to establish future work directions of this article: A) What is the relationship of the parameter H with

other multifractal measures such as the Hurst exponent generalized or the generalized fractal dimension? B) What relationship exists between the trace moment (TM) or double trace moment (DTM) method with other methods with which multifractal exponents are estimated, for example multifractal detrended fluctuation analysis (MF-DFA), total triangle areas (TTA), generalized Hurst exponent (GHE), among others (see for a review `https://doi.org/10.1016/j.physa.2021.126288`)?

As rightly pointed out, the exponent denoted '$H$' in the Universal Multifractal framework (like this paper) characterizes the degree of conservation of the mean field across scales ($H > 0$ specifying growth with scale and $H < 0$ decrease). With respect to Eq. 8, $H$ corresponds to the order of fractional integration required to get $\phi_\lambda$ from $\varepsilon_\lambda$. A reference to (Tessier et al., 1993) is added in updated text now, for better understanding with spectral slope $\beta$ to which the connection of $H$ is used in UM context. A greater $H$ corresponds to stronger long range correlation (see Eq. 9).

A) The UM parameter $H$ is not identical to the classical Hurst exponent, which in any case has undergone a number of modifications/generalisations. But both quantify long range correlations for $H > 0$. This is clarified in the text to avoid confusion. We also emphasise that multifractality requires more than a scaling exponent to be statistically characterised, contrary to uni/mono-fractals.

B) UM is one of the few multifractal frameworks whose foundations are explicitly stochastic and to be applicable to space-time fields, not only to time processes. These important and convenient features motivated our choice. However, and in response to the referee's stimulating questions, this does not prevent us to translate in the revised version the main results using descriptors of other formalisms, including those suggested by the referee (e.g. the generalised Hurst exponents $H_q$ in relation to the UM $K(q)$), as far as possible and pointing out any limitations. For instance, deterministic frameworks restrict themselves above extremes, contrary to stochastic frameworks and therefore there is only limited overlap between the range of their fluctuations.

3. Section 3.2 establishes a way to understand the effect of a higher threshold in the Universal Multifractals framework. The authors could highlight the difference of this method with others where the bias introduced in the estimation of multifractal exponents is considered (see for example `https://doi.org/10.1016/S0378-4371(96)00165-3` and `https://doi.org/10.1103/PhysRevE.95.042311`).

This paper focuses on the parameter estimation biases associated with an upper bound on the data. To our knowledge, it is a novel issue not addressed in the literature so far. This was achieved in the specific framework of UM analysis. However, this is very likely to also affect other types of analysis. In conclusion, we will point out what can be deduced from the relationships revealed with various formalisms and what remains to be done for future work.

*Minor issues*

75  1. Eq. (4), (5), (6), (7) and (8), do not have explicit references from which they were taken before being placed as was done for Eq. (3). The above, although it is a minor change, is suggested so that those readers who do not know much about the Universal Multifractals approach can inquire about it, and therefore, for the article to have a greater scope.

80  The main reference for Eq. (4), (5), (6) and (7) are Schertzer and Lovejoy (1987, 1988) which was referred at the start of the section. To make it more accessible to readers, we have added and reorganised them like Eq. (3). Also added the reference for Eq. (9) in same format which was missing earlier.

2. In equation (7) the restriction of $1/\alpha + 1/\alpha' = 1$ is not clear, it is suggested to clarify
85  this restriction and what $\alpha'$ represents in said equation.

The caption was not intended as a 'restriction' but as the definition of $\alpha'$, a sort of dual index of $\alpha$. The equivalent relation $\alpha' = \frac{\alpha}{\alpha-1}$ can be easily derived from i. We only followed the classical UM framework presentation.

3. In the title of Figure 5, panel (c), it is not clear the upper threshold condition placed
90  on the power available, the authors could revise this part or clarify this condition in words ("values>rated power = rated power").

For more clarity an additional line is now added to explain the operation done in brackets.

4. Section 3.2, which explains the effect of a higher threshold in the Universal Multi-
95  fractals framework, should go as one more subsection of section 2. The above, to give greater cohesion to the article since at this time the manuscript remains disconnected between the results obtained with the empirical data of the wind farm and the numerical simulations. Furthermore, it seems to me to be a new part of the article and one that deserves to be highlighted.

100  We understand the first part of your comment. It indeed is an extension of the methodology in section 2. However, since this is a new theoretical development that intents to explain the effect of threshold, we wanted to keep it separate from 'Methods' in Section 2 which is referring previous works. Following your remark, this was clarified in the manuscript.

105  To highlight it as new part of the article, we could add a new section (say section 4) solely dealing with theoretical development and numerical simulation. But that will increase total number of sections in article to 6 and leave current section 3.1 rather small to stand on its own.

5. In section 3.3 it is not entirely clear how each simulation of the discrete cascades was generated. Thus, it is necessary to establish in greater detail how each of these ensemble simulations were achieved

Thank you. We have added an extra line in caption of Fig. 7 to better convey each simulation. An additional line is added at start of the section 3.3 explaining briefly how the simulation is performed. Additional details and references could obviously be provided if the reviewer believes this is currently not enough.

**References**

Schertzer, D., Lovejoy, S., 1987. Physical modeling and analysis of rain and clouds by anisotropic scaling multiplicative processes. Journal of Geophysical Research: Atmospheres 92, 9693–9714. URL: https://agupubs.onlinelibrary.wiley.com/doi/abs/10.1029/JD092iD08p09693, doi:10.1029/JD092iD08p09693.

Schertzer, D., Lovejoy, S., 1988. Multifractal simulations and analysis of clouds by multiplicative processes. Atmospheric Research 21, 337–361. URL: http://www.sciencedirect.com/science/article/pii/016980958890035X, doi:10.1016/0169-8095(88)90035-X.

Tessier, Y., Lovejoy, S., Schertzer, D., 1993. Universal multifractals: Theory and observations for rain and clouds. Journal of Applied Meteorology and Climatology 32, 223 – 250. URL: https://journals.ametsoc.org/view/journals/apme/32/2/1520-0450_1993_032_0223_umtaof_2_0_co_2.xml, doi:10.1175/1520-0450(1993)032<0223:UMTAOF>2.0.CO;2.

---

## Author Comment (AC2)

**Response to comments from editors and reviewers**

Part 1: Multifractal analysis of wind turbine power and the associated biases (npg-2024-5)

preprint in Nonlinear Processes in Geophysics from 02 Feb 2024

⁵ Editor's decision received on
* * *
The authors would like to thank the reviewer for evaluating the paper and providing a detailed feedback. We appreciate the positive feedback and have tried our best to incorpo-

¹⁰ rate them in the updated manuscript. Hopefully the modifications are in accordance with the reviewer's expectations and quality of the journal.

Please find below our point-by-point response to the comments (Reviewer comments are shown in black and author responses are in blue).

**Anonymous Referee #2, 25 Mar 2024**

¹⁵ This work arises from the difficulty of analysing by means of a multifractal approach the empirical turbine power, since the production of wind turbines is limited by a maximum or nominal power. An adequate theoretical framework of wind behaviour is crucial for understanding its effect on wind turbine power. On the other hand, it is known that the existence of instrumental limits causes biases.

²⁰ In this work, authors employ the universal multifractal formalism for characterizing the small-scale fluctuation in wind power production and for highlighting the aforementioned biases and their influence on statistical analysis of turbine power.

To my mind, the work is very pertinent. It is very well argued. The objectives are achieved with a robust methodology, and thus my recommendation is to accept the paper.

²⁵ Some minor suggestions are made below:

Thank you very much for taking the time to read and review our manuscript. We greatly appreciate the very positive feedback and encouragement. Please find our response to the points raised below..

³⁰ • On line 127, page 5 appears that the sampling frequency is 15 s. It is suggested to clearly indicate if this is the sampling resolution of the variables.

This is the sampling frequency of Turbine power and local wind velocity measured by turbine. The sampling frequency of latter is now specified in text.

Other variables were measured at a finer frequency than this, however since they
³⁵ were analysed along side $P_t$, the resolution of 15 s was kept for power available $P_a$. A line is added to clarify this when $P_a$ is introduced.

- One of the main objectives of this work is to analyse several biases due to wind measurements. On line 130, page 6 the text reads: 'There are instances where the turbine failed to produce any power and had to consume energy for its basic operation. This results in negative values in data, and for realistic analysis, they were considered as zero'. It could be interesting if the authors could provide some light about the effect of applying this criterion.

  This is indeed an aspect we try to address. In current output based analysis using UM which in currently used format deals with positive fields, interpretation of the exact effect is out of scope. We have tried to address this indirectly by considering those negative readings as zero whose effect is rather well known (Gires et al., 2012). However, for the theoretical aspect only the effect of upper threshold is considered here to avoid complexity, and because it was not yet addressed in the literature. Following your remark, this is clarified in the manuscript now.

- On line 172, page 7 add a space between the full stop and the sentence that initiates as 'For a conservative...'. The same is said on line 246, page 12.

  Thank you. Updated both places.

- On line 207, page 9. It is suggested to clarify what are the parameters $q_s$ and $q_D$.

  The parameters are expanded to convey exact meaning.

- Figure 4. Please replace the current format of the date in the title of the Figures (2021_05_00_00_00__2021_05_26_23_59_30) with a more understandable date format.

  Thank you. The figures are now updated with a more legible format.

**References**

Gires, A., Tchiguirinskaia, I., Schertzer, D., Lovejoy, S., 2012. Influence of the zero-rainfall on the assessment of the multifractal parameters. Advances in Water Resources 45, 13–25. URL: https://www.sciencedirect.com/science/article/pii/S0309170812000814, doi:https://doi.org/10.1016/j.advwatres.2012.03.026. space-Time Precipitation from Urban Scale to Global Change.